# A Comparison of the Chronologies of Introduced versus Native Coniferous Tree Species Growing in Northwestern Poland during the Period of Global Warming

Anna Cedro [1,*] and Grzegorz Nowak [2]

1   Institute of Marine and Environmental Sciences, University of Szczecin, Mickiewicza 16, 70-383 Szczecin, Poland
2   Laboratory of Dendrology and Green Area Management, Department of Landscape Architecture, West Pomeranian University of Technology in Szczecin, Słowackiego 17, 71-434 Szczecin, Poland; grzegorz.nowak@zut.edu.pl
*   Correspondence: anna.cedro@usz.edu.pl

**Abstract:** The ongoing climatic changes are causing the extinction of numerous species or their withdrawal from previously occupied areas. The environmental and economic significance of introduced species may increase. The aim of the present study was to examine the rate of growth of coniferous species growing in northwestern Poland and to analyze the tree ring width–climate relationships. Six tree species were selected for this study. Two of these species have natural occurrences in Poland: *Pinus sylvestris* and *Larix decidua*. The remaining four species were introduced from North America: *Chamaecyparis lawsoniana*, *Thuja plicata*, *Pseudotsuga menziesii*, and *Pinus strobus*. Samples were collected from 131 trees using a Pressler borer at 1.3 m above ground. Tree ring widths were measured down to 0.01 mm. Climatic data were retrieved from a weather station located 23 km from the study plot. The average tree ring width reaches the lowest value for the *P. sylvestris* chronology (1.62 mm/year) and for *P. strobus* (1.69 mm/year), and the highest value is reached for *T. plicata* (2.80 mm/year) and *P. menziesii* (2.56 mm/year). The analysis of weather conditions in the designated pointer years and the response function analysis indicate that winter and early spring air temperature is the factor responsible for the formation of wide tree rings in the following species studied: *P. sylvestris*, *C. lawsoniana*, *P. menziesii*, and *T. plicata*. For *L. decidua* and *P. strobus*, the climate–growth relationships are different: weather conditions in the previous growth year are important, and it is the weather in the late spring and summer months. Two of the investigated introduced species (*T. plicata* and *P. menziesii*) are characterized by very good acclimatization and are best adapted to the new habitat during the current climate changes. These tree species can constitute a basis for replacing native species, which, due to increasingly severe droughts and higher temperatures, are doing less and less well in their current habitats. Foresters wanting to conduct sustainable forest management will look for replacement species that are well adapted to new habitat conditions in order to maintain the continuity of forest cover.

**Keywords:** sustainable forestry; Scots pine (*Pinus sylvestris*); common larch (*Larix decidua*); Lawson cypress (*Chamaecyparis lawsoniana*); western red cedar (*Thuja plicata*); Douglas fir (*Pseudotsuga menziesii*); eastern white pine (*Pinus strobus*); dendrochronology; dendroclimatology

## 1. Introduction

Climate changes resulting from global warming and human activity pose a challenge to foresters, whose priority is to conserve forests and trees growing therein. One of these activities is the introduction of alien tree species, which is a centuries-old tradition of practical forest breeding [1]. Forests play a key part in the economy of Poland and represent important leisure and recreation sites. Climate changes are causing Europe-wide economic consequences [2]. These threats may result in shifts to natural occurrence ranges

of economically important species, changes in the taxonomic composition of forests in favor of deciduous species, appearance and increased activity of new pathogens, reduced forest stand health and stability (especially pine and spruce stands), and gradations of invasive plant and animal species [2–5]. The use of introduced trees, often free from pathogens and pests in the new habitat, enables adaptation to threats resulting from environmental changes. However, one should remember the possible harmful effects resulting from the possible invasiveness of plants and not use solid stands of alien species in place of native species [6]. Such rapid changes are causing the extinction of numerous species or their withdrawal from previously occupied areas. Under such circumstances, previously undervalued plant or animal taxa, including uncommon or introduced taxa, may gain ecological or economic significance. To monitor climate changes and their impact on forest ecosystems, the results of dendroclimatological research may be used [7–16]. Many of the introduced tree species are larger, grow faster, and have better wood value than European species [15]. We often cannot predict the ecological consequences of introducing alien species into forests, and they may not become apparent even after several dozen years, which may threaten natural forest ecosystems [2]. It is, therefore, significant to keep expanding the base of knowledge on the ecology of all taxa, especially those that retain high immunity and resilience in the face of changing environmental conditions. Experimental forest plots, for instance, in western and northern Poland, were established as early as the 19th century by the German forester Schwappach [16–18]. Some of them survive to the present day, which enables performing research on various species of introduced plants. Information obtained from such old forest stands is used in sustainable forestry. They allow conclusions to be drawn regarding tree productivity, resistance to abiotic and biotic factors occurring in new habitats, and acclimatization. These results are the basis for appropriate forest management, restoration, protection, preservation of biodiversity, and the simultaneous use of many forest services.

The present study aimed to (i) study the growth rate of six species of coniferous trees growing in northwestern Poland, (ii) draw comparisons between local chronologies of this species, (iii) analyze the tree ring width–climate relationships in this species growing within the same habitat and climatic conditions, and (iv) evaluate tree features.

## 2. Materials and Methods

### 2.1. Study Area

For the present study, we selected six species of coniferous trees growing in northwestern Poland on state-owned forest land (Figure 1).

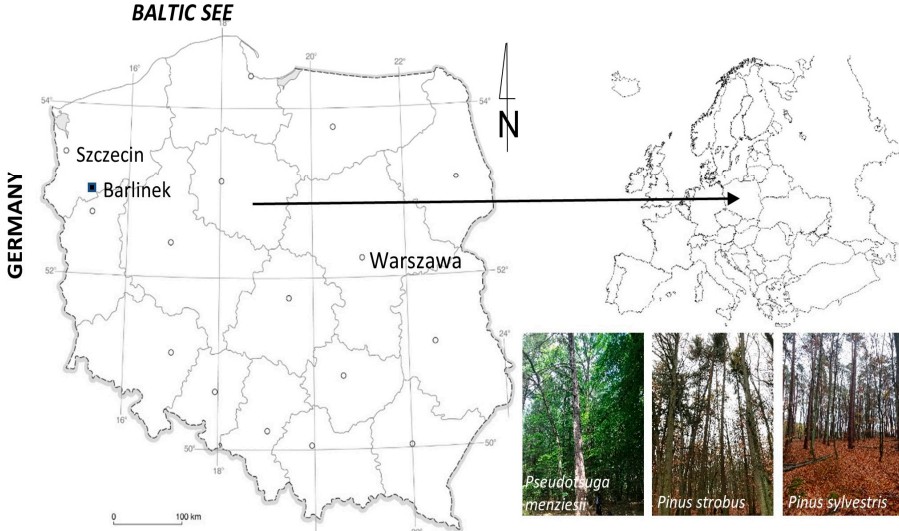

**Figure 1.** Location of the research site.

The area of the Barlinek Forest Inspectorate was shaped by the last glaciation. It rises gradually from outwash plain lowlands in the southwest toward the northeast, where the relief is more diverse, with numerous morainic elevations and ribbon lakes. The end moraine area within the limits of the Barlinek Forest Inspectorate is incised meridionally by the tunnel valley of the Płonia River. The substratum within this area is composed of outwash sands (ca. 71%), glacial sands (ca. 18%), peats and mucks superjacent to peats (ca. 2.5%), and moderately to highly sandy tills (ca. 1.5%) [19–21]. There are numerous lakes within the limits of the Forest Inspectorate, alongside smaller basins and forest ponds. In addition to playing a biocenotic part, these are also involved in the so-called small water retention and in stabilizing the groundwater level [22]. Fresh mixed deciduous forest is the dominant habitat type, occupying 67% of the forest area. Fresh forest occupies 17%, and fresh mixed coniferous forest occupies 13% of the forest area. The area percentage of the main forest-forming species equals 67% for pine, 14% for beech, and 10% for oak.

### 2.2. Climate Data

For dendroclimatic analyses, we utilized data retrieved from the IMGW weather station in Gorzów Wielkopolski (station no. 12300), located 23 km south of the study plot. The data series included air temperature and precipitation totals spanning 1948–2019 (72 years) and sunshine duration spanning 1965–2019 (55 years) (Figure 2).

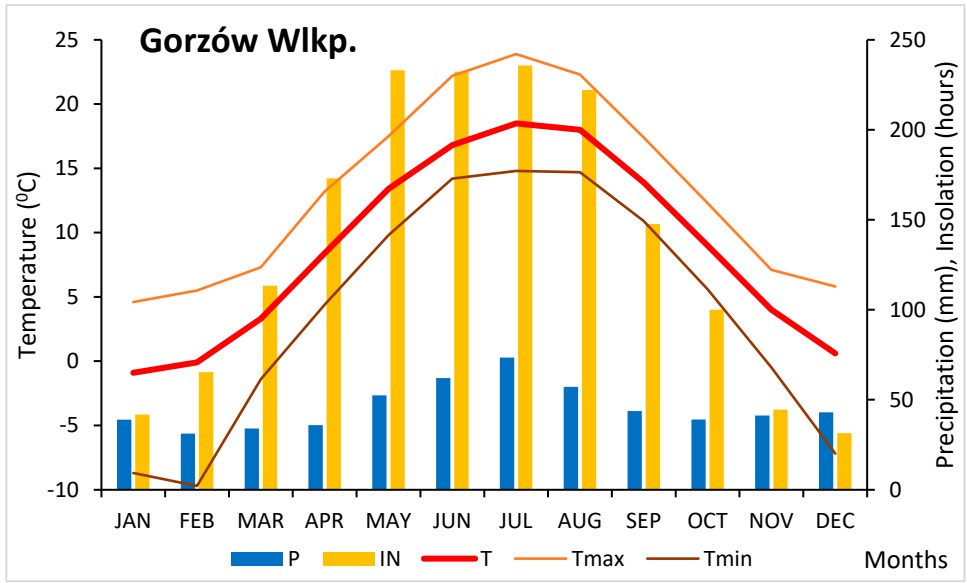

**Figure 2.** Mean monthly air temperature (T), mean maximum temperature (Tmax), mean minimum temperature (Tmin), mean monthly precipitation (P) for 1948–2019 (72 years), and mean monthly sunshine duration (IN) for 1965–2019 (55 years)—data from the weather station in Gorzów Wlkp. (no. 12300).

The mean annual air temperature for this period equals 8.8 °C. The year 1956 was the coldest year, with a mean temperature equal to 6.7 °C. The year 2019 was the warmest year, with a mean temperature of 10.9 °C. In each year in the dataset, July is the warmest month, with an average air temperature equal to 18.5 °C (ranging from 14.8 to 23.9 °C), and January is the coldest month, with a mean air temperature equal to −0.9 °C (ranging from −8.7 to 4.6 °C). The average rainfall sum equals 551 mm. In the driest years, rainfall totals drop to 337 mm, reaching 872 mm in the most humid years. In each year, the lowest precipitation totals are noted for February (31 mm), and the highest totals are found for July (73 mm). Precipitation varies through the summer months on a year-to-year basis. Rainfall shortages occur frequently (monthly totals below 30 mm), but high precipitation totals are also frequent (monthly sums above 120 mm). The annual insolation at the weather station in Gorzów Wielkopolski equals 1640 h (ranging from 1222 h in 1966 to 2147 h in

2018). The lowest insolation is noted for January (monthly mean: 42 h), November (44 h), and December (31 h). Insolation values higher than 200 h per month are observed in May (on average, 233 h), July (232 h), and August (222 h). Snow cover occurs in this region for less than 30 days. Most frequently, it appears after 20 December and disappears before 10 March, and it is thin and unstable. Growing season lasts, on average, 235 days (23 March through 11 November) [23].

According to the scheme of climate zones and subzones by Heinze and Schreiber [24], this area is located in subzone 7a, with an average long-term minimum temperature from −17.7 to −15.0 °C. The climate of this area is classified in the Köppen–Geiger classification as Dfb (cold climate, without a dry season and warm summer) [25].

*2.3. Characteristics of the Selected Tree Species*

Six species of coniferous trees occurring within the Forest Inspectorate in sufficient abundance to perform dendroclimatic examination. These included two species having their natural stands in Poland: the Scots pine *Pinus sylvestris* L. (PS) and the common larch *Larix decidua* Mill. (LD). Further, four species were introduced from North America: the Lawson cypress *Chamaecyparis lawsoniana* (A. Murray bis) Parl. (CL), the western red cedar *Thuja plicata* Donn ex D. Don (TP), the Douglas fir *Pseudotsuga menziesii* (Mirb.) Franco (PM), and the eastern white pine *Pinus strobus* L. (PST).

The *P. sylvestris* is the basic forest-forming species in Poland, and it has a high economic significance. It reaches a height of up to 30–40 m. It occurs in Europe and Asia, as far east as northeastern China and the Sea of Okhotsk coast. It forms pine forests that are dry, fresh, humid, marshy, and mixed [26,27]. The *L. decidua* reaches a height of up to 40–50 m. It occurs in Western and Central Europe, usually in the mountains, where it forms forests together with the Swiss pine, with an admixture of the European spruce *Picea abies*. In lowland forests, it is an introduced component [27].

The *C. lawsoniana* was imported to Europe (England) in 1854. In Poland, its presence was recorded for the first time in 1870 in the Kórnik arboretum [28]. It is native to the region of North America, located at the junction of California and Oregon, within a humid, oceanic climate, where it grows up to 40–50 m high. In the wild, it forms forests together with *Abies grandis*, *Picea sitchensis*, *P. menziesii*, *T. plicata*, and *Tsuga heterophylla* [27,29]. The *T. plicata* was introduced in England in 1853, and it was recorded in the Kórnik arboretum [28] as early as 1861. In the wild, it reaches a height of 60–70 m. It occurs in western North America, mainly in the United States. It prefers a mild climate and humid soils along river banks, where it grows together with, e.g., *T. hetrophylla*, *P. sitchensis*, *P. menziesii*, and *Abies grandis* [27,29]. The *P. menziesii* was introduced in Europe (England) in 1827. In Poland, it had already been recorded in 1833 in the Niedźwiedź Garden in the vicinity of Kraków [28]. In the wild, it occurs along the Pacific coast, from southwestern Canada to northern California, either in monospecific forests or, e.g., with *Abies grandis*, *T. heterophylla*, and *T. plicata*. It is one of the highest trees in America, reaching up to 60–130 m [27,29]. The *P. strobus* has been noted in Europe since the 16th century (France). It was brought to Poland around 1798 (Chrzelice Garden) [28]. It is native to northeastern North America (Canada), where it grows up to 30–65 m high, and may grow together with, e.g., *Pinus resinosa*, *Tsuga canadensis*, or *Quercus alba* [27,29].

*2.4. Tree Ring Data Analysis*

Fieldwork was carried out in July 2020 (*C. lawsoniana*, *T. plicata*), June 2023 (*L. decidua*, *P. menziesii)*, and November 2023 (*P. sylvestris*, *P. strobus*). Tree height was measured using a Nikon Forestry Pro II laser rangefinder. The trunk diameter was measured using a medium gauge at a height of 1.3 m above ground level. Sampling for dendrochronological analyses was performed on the healthiest trees, i.e., those with no apparent trunk damage. Samples were taken from 131 trees using a Pressler borer at a height of 1.3 m above ground level (Table 1). In the laboratory, the samples were glued to boards, dried, and cut with a knife to obtain a clear view of the tree ring boundaries. Tree ring width (TRW) was measured

under a stereoscopic microscope to 0.01 mm using LBD_Measure software (version 1.0) [30]. Cross-dating between the individual tree TRW time series was performed using on-screen visual comparisons (high visual similarity) and statistical parameters commonly used as cross-dating coefficients in dendrochronology: Student's t-test, r correlation coefficient, and Gleichläufigkeit (GL%) [31,32]. To check the annual variability of the TRW, standard deviation (STD), mean sensitivity (MS), and autocorrelation coefficient lagged by one year (AC) were also calculated. TRW raw series from the site were averaged to tree ring width raw chronology. Trees with the lowest values of statistical parameters (t < 3.5, r < 0.5, and GL < 50%) were removed from the dataset. Only TRW raw chronologies that were successfully cross-dated and yielded a good match among themselves were selected for further analysis [33–35]. To remove the influence of the long-term age trend and disturbances of other environmental factors, TRW raw time series were standardized individually using ARSTAN program [36], using a two-phase detrending technique by fitting either a modified negative exponential curve or a regression line with a negative or zero slope). Residual site chronologies (RES) were derived by averaging the individual TRW series from each site, which were previously detrended and had autocorrelation removed [37]. The EPS coefficient (which describes the intensity of the common climate information between chronologies) was also computed [38].

**Table 1.** List of research plots with basic information.

| Lab. Code | Name | Species | Height of Trees (m) | Diameter at Breast Height Mean (min.–max.), (cm) | No. of Trees | No. of Samples | No. of Tree Rings |
|---|---|---|---|---|---|---|---|
| PS | Scots pine | *Pinus sylvestris* | 23.5 | 48 (36–60) | 25 | 28 | 3339 |
| LD | Common larch | *Larix decidua* | 31.0 | 63 (38–96) | 24 | 24 | 2560 |
| CL | Lawson cypress | *Chamaecyparis lawsoniana* | 22.0 | 36 (27–51) | 21 | 37 | 3727 |
| TP | Western red cedar | *Thuja plicata* | 34.0 | 61 (35–105) | 22 | 25 | 2662 |
| PM | Douglas fir | *Pseudotsuga menziesii* | 34.0 | 66 (44–99) | 25 | 25 | 2335 |
| PST | Eastern white pine | *Pinus strobus* | 30.0 | 47 (33–60) | 14 | 19 | 1691 |
| | | | | $\Sigma$ | 131 | 158 | 16,314 |

To study the growth–climate relationship, correlation and response function analysis and the analysis of pointer years were employed. Average monthly air temperature (T), monthly rainfall (P), and monthly insolation (IN) from June of the year preceding growth (pVI) to September of the growing year (IX) were used for correlation and response function analysis. The analysis was performed separately for temperature, precipitation, and insolation, obtaining $r^2$ values (regression coefficients of determination) for each climate parameter [34,39,40]. The analysis of pointer years was carried out using the TCS program [41], calculating the interval trend: positive years (+), characterized by an increase in the width of the rings (t = 1) compared to the previous year (t − 1), and negative years (-), with a decrease in the width of the rings (t = 1) compared to the previous year (t − 1) [32,42]. A given year was considered a pointer year (positive or negative) if the interval trend exceeded the critical threshold—90% for a minimum of 10 trees.

## 3. Results

### 3.1. Tree Features

The *P. sylvestris* and the *C. lawsoniana* species grow in clusters, and the remaining species are scattered over larger areas, together with other species. The *T. plicata* and the *P. menziesii* trees reach the tallest heights (34 m). The *L. decidua* and the *P. strobus* species display slightly lower heights (31 and 30 m, respectively). The lowest heights were reached by the *P. sylvestris* (23.5 m) and the *C. lawsoniana* (22 m) trees. The *P. menziesii* trees are distinguished by the highest average trunk diameter (66 cm, ranging from 44 to 99 cm). The lowest trunk diameter was observed for *C. lawsoniana*: on average, 36 cm, ranging from 27 to 51 cm (Table 1). According to Kraft's classification, assessing the structure of the crown and the biological position of the plant in the population [43], *P. sylvestris* trees belong to the main, dominant stand, class 1. The dominant trees of *C. lawsoniana* belong to class 2, and the co-dominant trees of *L. decidua*, *T. plicata*, and *P. menziesii* belong to class 3. *P. strobus* belongs to a subordinate stand, class 4.

The majority of trees display good health, and the natural branch snag is noticeable. The *T. plicata* trees are the most healthy, as no trunk damage or cavities were observed. In the case of the *C. lawsoniana* and the *P. sylvestris*, we observed no mechanical damage or wood decay, only strong reductions in tree ring width, especially conspicuous in recent years. The *P. menziesii* trees appear entirely healthy, but in five cases, we recorded the occurrence of wood rot at the coring spot. The poorest health was displayed by one native species, *L. decidua*, and one introduced species, *P. strobus*. The *L. decidua* trees had weak crowns, and four trees had rotten trunks. In the case of *P. strobus*, we noted two withered trees and a further three trees had largely rotten trunks. Leaving dead trees and stumps in the forest is very important for the biodiversity of forest plant communities, contributing to the sustainable development of forest cultivation [17]. During the research, young plants of *T. plicata* and *C. lawsoniana* were observed, which indicates their ability to penetrate the surroundings and regenerate naturally. Currently, the seedlings do not pose a threat to the native flora because in unfenced areas, the plants are eagerly browsed by forest animals, and the relatively small number of trees does not exert pressure on the surrounding tree stands.

### 3.2. Tree Ring Width Chronologies

A chronology was compiled for each studied species. The shortest chronology, *C. lawsoniana* (CL), spans 114 years from 1906 to 2019, and the longest chronologies span 136 years (from 1887 to 2022, *L. decidua*—LD chronology) and 135 years (from 1887 to 2023, *P. sylvestris*—PS chronology) (Table 2). On average, a chronology is based on 18 individual growth curves, ranging from 13 curves for the *P. strobus* (PST) chronology to 21 curves for the *L. decidua*—LD chronology. The average tree ring width is the lowest for the PS (*P. sylvestris*) chronology (1.62 mm/year) and for the PST (*P. strobus*) chronology (1.69 mm/year), while the highest value is recorded for the TP (*T. plicata*) chronology (2.80 mm/year). Cumulative values of tree ring widths reveal the following patterns: *L. decidua* (LD) reaches the highest tree ring widths through the first 20 years of life; *T. plicata* (TP) and *P. menziesii* (PM) reach the highest tree ring widths between 20 and 40 years of life; and the TP chronology displays the highest tree ring widths above 40 years of life. PST has the narrowest tree rings through the entire period of tree life. Further, from the age of about 110 years, PS (*P. sylvestris*) and CL (*C. lawsoniana*) attain minimum growth values (Figure 3). The EPS value remains >0.85 for all chronologies from 1946 to 2019 (Table 2).

The most convergent chronology pair, as expressed by the t index, are PM/TP chronologies (t = 11.21). High t values (>7.0) are also obtained for the following pairs of chronologies: CL/PM and PS/PST. The lowest t index values are observed for the CL/PST chronologies (2.30). Low t index values (t < 3.0) are also displayed by the pairs CL/LD and LD/PS (Table 3). As expressed by the GL index, the PS and PST chronologies represent the most convergent chronology pair (73%). GL values > 65% are also obtained for the following pairs of chronologies: LD/PS, CL/PS, PM/PS, CL/PM, and PM/PST. The lowest GL value

was obtained for the CL/LD pair (44%). Comparably low GL values (<50%) were noted for the pairs of chronologies: LD/PST, CL/TP, and PST/TP (Table 3).

**Table 2.** Basic statistics of measured and index (residual) coniferous local chronologies: *P. sylvestris* (PS), *L. decidua* (LD), *C. lawsoniana* (CL), *T. plicata* (TP), *P. menziesii* (PM), and *P. strobus* (PST). Abbreviations: TRW—tree ring width; SD—standard deviation; AC—first order autocorrelation; MS—mean sensitivity; EPS—expressed population signal; rbt—series intercorrelation.

| Lab. Code | No. of Years | Time Span | No. of Samples | Mean TRW (min.–max.) (mm) | Measured Chronologies | | | Indexed Chronologies | | | EPS >0.85 | rbt |
|---|---|---|---|---|---|---|---|---|---|---|---|---|
| | | | | | SD | AC | MS | SD | AC | MS | | |
| PS | 135 | 1889–2023 | 19 | 1.62 (0.91–2.02) | 0.922 | 0.753 | 0.323 | 0.239 | −0.114 | 0.302 | 1896–2023 | 0.644 |
| LD | 136 | 1887–2022 | 21 | 2.21 (1.25–4.65) | 1.082 | 0.643 | 0.374 | 0.257 | −0.088 | 0.310 | 1902–2022 | 0.640 |
| CL | 114 | 1906–2019 | 16 | 1.76 (1.26–2.18) | 1.371 | 0.763 | 0.456 | 0.323 | −0.050 | 0.418 | 1914–2019 | 0.665 |
| TP | 127 | 1894–2020 | 19 | 2.80 (1.46–4.80) | 1.695 | 0.625 | 0.381 | 0.314 | 0.151 | 0.311 | 1911–2020 | 0.671 |
| PM | 119 | 1904–2022 | 17 | 2.56 (1.34–3.42) | 1.357 | 0.742 | 0.328 | 0.241 | 0.020 | 0.293 | 1923–2022 | 0.654 |
| PST | 125 | 1899–2023 | 13 | 1.69 (0.75–2.75) | 0.950 | 0.676 | 0.328 | 0.229 | 0.012 | 0.2642 | 1946–2023 | 0.536 |

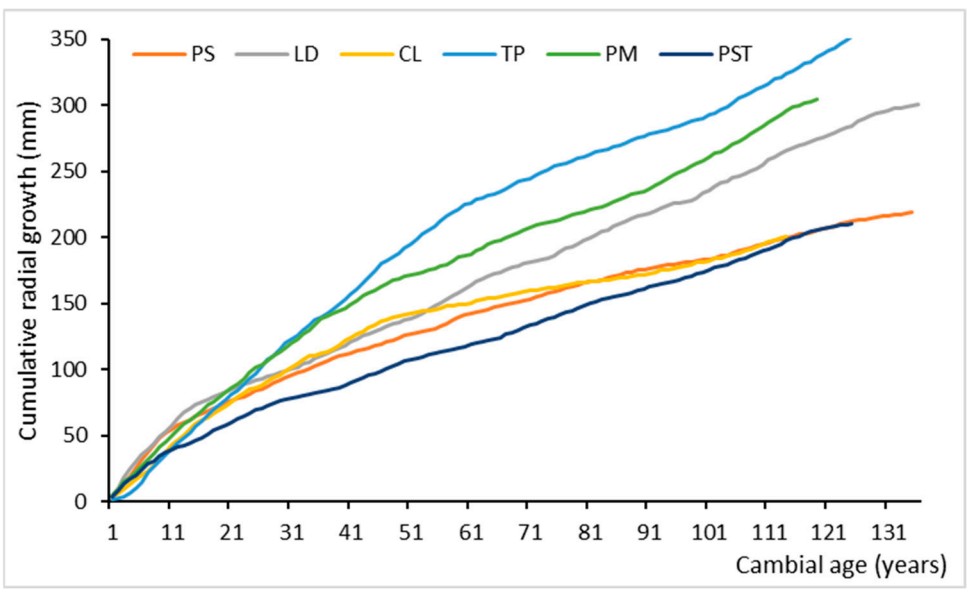

**Figure 3.** Cumulative radial growth of *P. sylvestris* (PS), *L. decidua* (LD), *C. lawsoniana* (CL), *T. plicata* (TP), *P. menziesii* (PM), and *P. strobus* (PST).

**Table 3.** Similarity of local chronologies of *P. sylvestris* (PS), *L. decidua* (LD), *C. lawsoniana* (CL), *T. plicata* (TP), *P. menziesii* (PM), and *P. strobus* (PST), as measured with t and GL (%) values.

| t/GL | PS | LD | CL | TP | PM | PST |
|---|---|---|---|---|---|---|
| PS | X | 2.50 | 5.02 | 5.40 | 6.60 | 7.80 |
| LD | 67 | X | 2.40 | 3.83 | 3.28 | 3.07 |
| CL | 65 | 44 | X | 6.48 | 7.72 | 2.30 |
| TP | 51 | 53 | 47 | X | 11.21 | 3.78 |
| PM | 66 | 53 | 65 | 50 | X | 3.86 |
| PST | 73 | 47 | 59 | 45 | 69 | X |

### 3.3. Correlation and Response Function

The correlation analysis for *P. sylvestris* (PS) for air temperature (T) yielded negative coefficient values for September and October of the year preceding growth. Positive values were obtained for February. For precipitation (P), only a positive correlation was obtained for November of the preceding year. For insolation (IN), only one statistically significant correlation value was noted: positive for January. The highest $r^2$ was obtained for T = 34%, a lower value was obtained for IN = 21%, and the lowest was obtained for P = 19% (Figure 4).

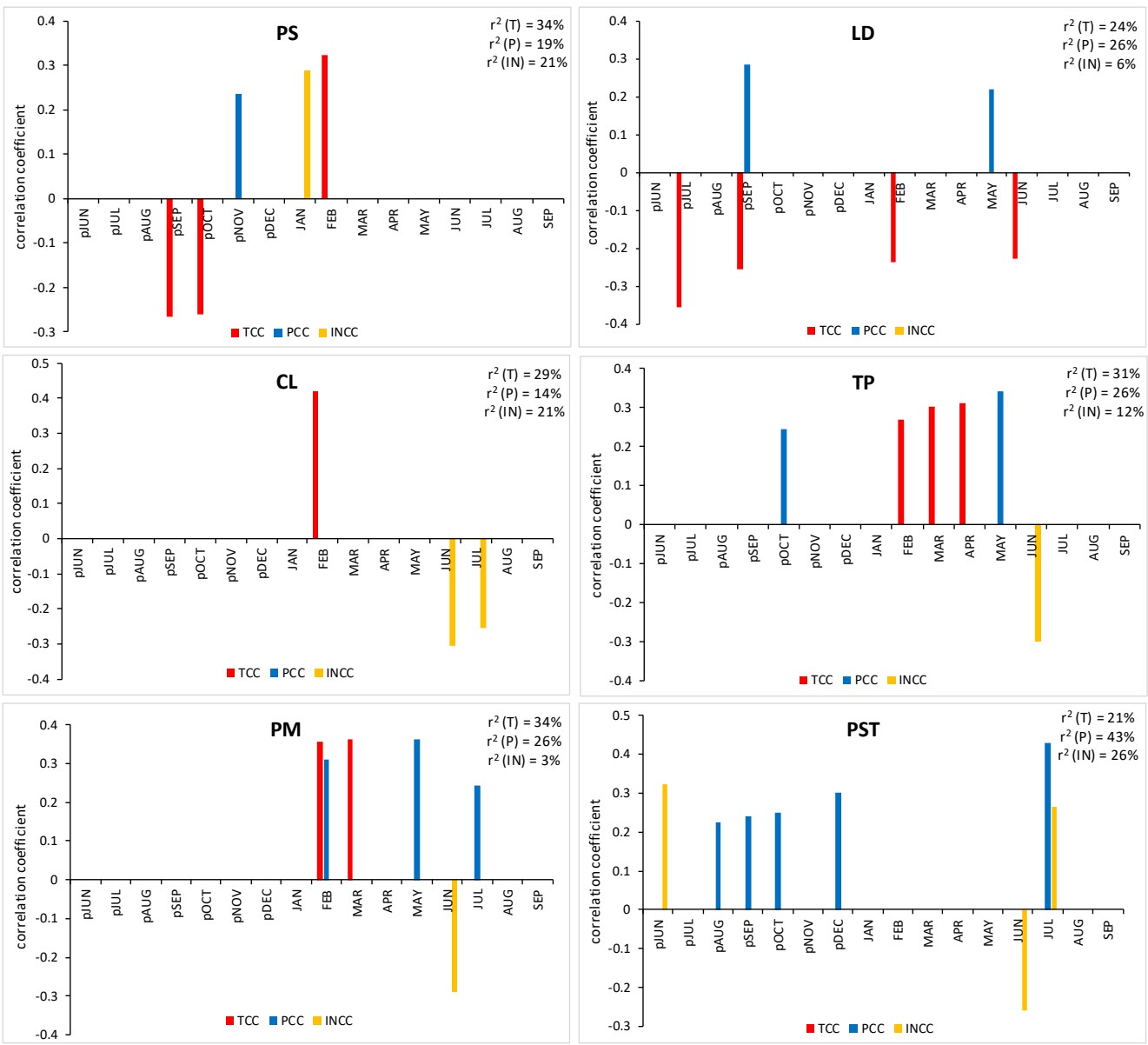

**Figure 4.** Results of correlation (CC) analyses for *P. sylvestris* (PS), *L. decidua* (LD), *C. lawsoniana* (CL), *T. plicata* (TP), *P. menziesii* (PM), and *P. strobus* (PST) chronologies for temperature (T), precipitation (P), and insolation (IN). Bars denote significant values ($p \leq 0.05$); p, preceding year; $r^2$, determination coefficient.

The analysis for another native species, *L. decidua* (LD), yielded only a negative correlation for T for the following months: July and September of the preceding year and February and June of the growth year. For P, the analysis yielded only positive statistically significant coefficient values for September of the preceding year and May of the growth year. For IN, no statistically significant values were obtained for the analyzed 16-month

period. The highest r$^2$ value was obtained for P = 26%, slightly lower for T = 24%, and very low for IN = 6% (Figure 4).

For *C. lawsoniana* (CL), the results are as follows: positive values for February of the growth year for T. For P, no statistically significant values were obtained for the analyzed 16-month period. For IN, negative values were obtained for June and July of the growth year. Regression determination coefficients attain a value of 29% for T, 21% for IN, and 14% for P (Figure 4).

For air temperature, the results of the response function analysis for *T. plicata* (TP) are as follows: positive values for February, March, and April of the growth year. Only positive values were obtained for precipitation: for October of the preceding year and for May of the growth year. For insolation, only one statistically significant value (−0.2982) was obtained for June of the growth year. The highest r$^2$ value was obtained for T = 31%, a lower value was obtained for P = 26%, and the lowest value was obtained for IN = 12% (Figure 4).

The analysis of *P. menziesii* (PM) yielded positive values of correlation for February and March. For precipitation, the statistical indices have positive values for February, May, and July of the growth year. For IN, there is also only one statistically significant negative value for June, similar to the TP chronology. The r$^2$ value is the highest for T and equals 34%; for P, it equals 26%; and for IN, it is the lowest at 3% (Figure 4).

The response function analysis for *P. strobus* (PST) yielded no statistically significant values for the analyzed 16-month period. For precipitation (P), all values are positive for the period spanning August–October and December of the preceding year and for July of the growth year. For insolation (IN), we noted both positive (pJUN, JUL) and negative values (JUN). The highest r$^2$ value was obtained for P (43%), a value of 26% was obtained for IN, and a value of 21% was obtained for T (Figure 4).

*3.4. Pointer Years*

Figure 5 shows the compilation of pointer years for six chronologies for the period 1948–2019. In various chronologies, the same calendar year may be either positive or negative. For this reason, in the analysis below, we include only those years in which growth reactions (tree ring width increase or reduction in most trees) are consistent in at least three out of six chronologies compiled for the species considered here. The following years were identified as positive pointer years (+): 1953 (for the chronologies PS, TP, PST), 1957 (CL, TP, PST), 1965 (CL, TP, PM), 1977 (LD, TP, PM, PST), 1980 (CL, TP, PM), 1984 (PS, CL, TP, PST), 1993 (LD, CL, PST), 2004 (PS, CL, TP, PST), and 2007 (LD, TP, PM) (Figure 5). An analysis of weather conditions in these years indicates that the air temperature of winter and early spring and precipitation totals in the summer months are the factors influencing the formation of wide tree rings. The mean annual air temperature is above or below average or oscillates close to the average. In the winter months (January and February), the air temperature is positive, and March is warm as well. In the summer months, the air temperature is close to average, or the summers are rather cool. The annual precipitation sum is normal or higher than average, and from May to August, the precipitation is close to or higher than average. The annual sunshine duration is lower than the multi-year average. The insolation is lower than usual, especially in the summer months (Table 4). The year 1977 may serve as an example of a positive pointer year. The mean annual air temperature for this year is 8.5 °C (compared to the multi-year mean value of 8.8 °C). January is rather cool, but the mean air temperature is close to the multi-year mean; February and March are warm. The summer (July and August) is rather cool. The annual precipitation sum is very high (692 mm/year, i.e., one of the four most humid years in our data series). In the growing season, precipitation is higher than average (very high rainfall in August), except for June. The insolation for 1977 equals just 1297 h (compared to the multi-year mean, which is equal to 1640 h). In the summer months (especially in July and August), the insolation is very low, 184 and 146 h, respectively.

| Year | PS | LD | CL | TP | PM | PST | Year | PS | LD | CL | TP | PM | PST |
|------|----|----|----|----|----|-----|------|----|----|----|----|----|-----|
| 1948 | + |   | - |   |   |   | 1985 | - | + |   |   |   |   |
| 1949 |   | - | + |   |   |   | 1986 |   |   | - | - |   |   |
| 1950 | - |   | - |   | - | - | 1987 |   |   |   |   |   | + |
| 1951 | + | - |   |   |   | + | 1988 |   | + |   | + |   |   |
| 1952 |   |   | - | - | - | - | 1989 |   |   |   | - |   |   |
| 1953 | + |   |   | + |   | + | 1990 | + | - | + |   |   |   |
| 1954 | - | - | - | - | - | - | 1991 | + |   |   |   |   |   |
| 1955 |   | + | + |   |   |   | 1992 |   | - | - | - | - | - |
| 1956 | - |   | - | - | - | - | 1993 |   | + | + |   |   | + |
| 1957 |   |   | + | + |   | + | 1995 |   | - | - |   |   |   |
| 1958 | - | + |   | - |   | - | 1996 | - | + | + | + |   |   |
| 1959 |   | - |   |   |   |   | 1997 | + |   | - |   | + |   |
| 1961 |   |   | + |   | + |   | 1998 |   | + | + |   | - | - |
| 1962 |   | + |   |   |   |   | 1999 | - |   | - |   |   | + |
| 1963 | - |   | - |   |   |   | 2000 |   | - | + | - | - |   |
| 1964 |   | - | - | - | - | + | 2001 |   |   | - |   |   |   |
| 1965 |   |   | + | + | + |   | 2002 | - |   |   |   |   |   |
| 1966 | - |   |   |   |   |   | 2003 |   | + |   | - | - |   |
| 1967 |   |   | + |   |   | + | 2004 | + |   | + | + |   | + |
| 1968 |   | + | - |   | - |   | 2005 | - | + | - |   |   | - |
| 1969 |   |   |   | - | - | - | 2006 |   | - |   | - | - | + |
| 1970 | - |   | - |   |   |   | 2007 |   | + |   | + | + |   |
| 1971 |   |   |   |   |   | + | 2008 |   |   | - | - |   |   |
| 1972 |   |   |   | + |   |   | 2009 |   |   | + | + |   |   |
| 1973 |   |   |   | - |   | - | 2010 |   |   | - |   |   |   |
| 1974 |   | - |   |   |   | + | 2011 |   |   | + |   |   |   |
| 1975 |   | + |   |   |   |   | 2012 |   | + | - | + | + |   |
| 1976 | - | - | - | - | - | - | 2013 | - | - | + | - | - | - |
| 1977 |   | + |   | + | + | + | 2014 |   |   | + |   |   |   |
| 1978 |   | - |   |   |   | - | 2015 | - | - | - | - | - | - |
| 1979 |   | + | - | - |   | + | 2016 | + |   |   |   |   |   |
| 1980 |   |   | + | + | + |   | 2017 | - | - | - |   |   | - |
| 1982 |   | - |   |   | - |   | 2018 |   | + |   |   |   |   |
| 1983 | - | + | - | - |   | - | 2019 |   | - |   |   | - | - |
| 1984 | + |   | + | + |   | + |      |   |   |   |   |   |   |

**Figure 5.** Pointer years for chronologies of *P. sylvestris* (PS), *L. decidua* (LD), *C. lawsoniana* (CL), *T. plicata* (TP), *P. menziesii* (PM), and *P. strobus* (PST) during the study period (1948–2019). + denotes positive pointer years; - denotes negative pointer years; blue and orange denote interspecies pointer years.

The following years were identified as negative pointer years: 1950 (for the chronologies PS, CL, PM, PST), 1952 (CL, TP, PM, PST), 1954 (all species), 1956 (PS, CL, TP, PM, PST), 1969 (TP, PM, PST), 1976 (all species), 1992 (LD, CL, TP, PM, PST), 2015 (all species), 2017 (PS, LD, CL, PST), and 2019 (LD, PM, PST) (Figure 5). The analysis of weather conditions in the designated years suggests that winter air temperature (low temperatures) and summer precipitation (droughts) influence tree ring widths. As in the case of positive pointer years, the mean annual air temperature may vary in negative pointer years. Consistently, however, very low mean monthly air temperature occurs in winter months and in early spring (January, February, or March). The summer is usually warm, and some months may

even be hot. The annual rainfall sum is usually lower than average, and rainfall shortages occur in the growing season, especially from May to August. The annual insolation is most frequently above average, and very high insolation sums are noted in the summer months (Table 4). The year 1976 may serve as an example of a negative pointer year. It is a cool year (mean annual air temperature equals 7.7 °C, compared to the multi-year mean value of 8.8 °C). The air temperature of January is close to average, but negative monthly means are noted for February and March. July and August are very warm. The annual precipitation sum is low (482 mm compared to a mean value equal to 551 mm/year). Rainfall shortages are noted from June to August. The annual insolation (1718 h) is higher than average (1640 h). Insolation is higher than average from May to August.

**Table 4.** Weather characteristics in pointer years.

| Pointer Years | | Characteristic Features of the Weather |
|---|---|---|
| Positive (+) | Negative (−) | |
| | 1950 | **frosty JAN**; warm for the rest of winter; cool APR; average temperature in summer; **drought in MAY, JUN, and AUG** |
| | 1952 | **frosty MAR**; very warm APR; average temperature in summer; **rainfall deficits in MAY, JUL, and AUG** |
| 1953 | | **warm winter**; warm JUN; average temperature in summer; **heavy rainfall in MAY, JUN, and JUL** |
| | 1954 | **frosty winter and cool spring**; dry year; **drought in MAY and JUN**; high rainfall in JUL |
| | 1956 | **very frosty FEB**; cool spring and summer; heavy rainfall in JUN and AUG; **dry JUL** |
| 1957 | | **warm winter**; average temperature in summer; APR–JUN was quite dry; **very high rainfall in JUL** |
| 1965 | | FEB and MAR were cool; summer months were quite cool; **high rainfall in summer; insolation below average, especially in summer months** |
| | 1969 | **winter and the beginning of spring were frosty**; average temperature in summer; dry year; **drought in JUL and SEP**; annual sum of insolation below average; **low values of insolation, especially in MAY and AUG** |
| | 1976 | **very cold winter and spring**; warm JUL and hot AUG; dry year; **rainfall deficiency in summer; high insolation from MAY to AUG** |
| 1977 | | **warm winter and spring**; summer rather cool; **high rainfall in summer; very low insolation, especially in the summer months** |
| 1980 | | quite cold winter; cool summer; **very humid JUN; below average insolation, especially in summer** |
| 1984 | | winter and the beginning of spring were quite cold; cool summer; **high rainfall in MAY; JUN, and JUL; insolation below average, especially in JUL and AUG** |

As part of the analysis, we also identified those years in which growth reactions were inconsistent among the six studied species of coniferous trees. These include 1958, 1964, 1979, 1983, 1996, 1998, 2000, 2005, 2006, 2012, and 2013. These years include both positive and negative years among individual species. The tree ring width is influenced by various meteorological and habitat factors in these years (e.g., insect gradations, culturing procedures, etc.).

## 4. Discussion

For the main native coniferous species that forms most of the tree stands in Poland, *P. sylvestris* (PS), similar growth–climate relationships are observed as those obtained by other researchers from the south coast of the Baltic Sea. The results confirm the sensitivity of this species to low temperatures in winter/early spring and to rainfall deficits in the

summer season [7,8,44–49]. The mechanism of the relationship in February and March has not been fully explained, but the following climate changes (mainly warmer winters and earlier spring) should have a positive impact on the tree ring width for pine [47]. Also, the pointer years designated for this species mostly overlap and are determined by the same weather conditions [7,44,49]. The studied trees are also sensitive to thermal conditions in the autumn of the previous growing season, which is also confirmed by other studies [47,49,50].

The growth–climate relationships are different for the second analyzed native species, *L. decidua* (LD). Thermal and pluvial conditions of the previous growing season (July, September), winter (February), and spring and summer (May–June) are important for trees of this species. High rainfall and low temperatures in these months favor the formation of wide tree rings of larch. Koprowski [51] reported similar tree ring width–climate relationships for twelve larch chronologies from lowland Poland, i.e., a positive influence of high rainfall sums (also no negative relationships) and a negative influence of high air temperatures in the preceding and current growth seasons. In the larch trees from southern Poland, the sunshine duration in the previous October, insolation and air temperature in February and March, insolation in June and July, and precipitation in August and September are significant [52]. The results obtained by Wilczyński and Wertz [53] also point to weather conditions in the previous September and March of the growth year. The larch trees in the Polish part of the Carpathians are sensitive to the temperature in May (positive relationships) and precipitation in July (positive influence) and are negatively influenced by rainfall in April and negatively influenced by air temperatures in the previous May and July [54,55]. There is a positive influence of the temperatures in May and negative relationships with air temperatures of the preceding summer, and a negative influence of summer droughts is also noted in the Polish part of the Sudetes [56].

Three of the introduced species, *C. lawsoniana* (CL), *T. plicata* (TP), and *P. menziesii* (PM), have the same dominant climatic factor (air temperature) and the same period of dependence (February, March, and April of the year of growth formation) influencing the tree ring width. Higher than average temperatures in these months (warm winter and the beginning of spring) cause the formation of wide tree rings in the upcoming growth season in the studied trees. The positive correlation values for air temperature may be accompanied by positive values for precipitation (PM). In this period, an influx of warm and humid polar marine air masses from over the Atlantic favors tree health and high cambial activity. The summer month weather conditions are an additional factor influencing the tree ring width in the above species (wide tree ring formation is favored by lower than usual air temperatures, high precipitation, and low sunshine duration: CL, TP, and PM.

*C. lawsoniana* is rarely grown in the forests of Poland, which is why no data are available on the tree ring width–climate relationship in this species. The Sawara cypress (*C. pisifera*) growing in various parts of Poland is sensitive to air temperatures of the late winter/early spring period (February–March) and to precipitation sums in the summer months. Also, the regional pointer years designated for *C. pisifera* [57] are largely consistent with the years designated for *C. lawsoniana* (CL).

*T. plicata* is rarely subject to tree ring width–climate relationship studies in Poland due to its infrequent occurrence in forests and minor economic significance. Previous studies focused on two stands of this species (one in NW Poland and the other in the central part of the country). These studies found that air temperatures and precipitation in the late winter/early spring period (February–March) played a key part in shaping the tree ring widths [11,58]. A positive influence of precipitation in the previous November was also found, but no relationships in the growth season [11]. Gwenda and Koprowski [58] reported that precipitation in July had a positive influence. In another species of thuja (*T. occidentalis*), studies on the impact of drought on the condition of trees showed short-term and long-term components that were influenced by the severity and duration of the water stress [59].

The Douglas fir (*P. menziesii*) is the most frequently planted non-native conifer species in Poland. As it occurs frequently in various regions and its age often exceeds 100 years, it is a

frequent subject of dendrochronological and dendroclimatological studies [7,48,60–62]. The tree ring width–climate relationships for this species are similar to those for *P. sylvestris*, i.e., the dominant part in tree ring shaping is played by the late winter/early spring air temperature (positive relationships) [7,48,62]. The pointer years designated by Cedro [7] and Feliksik and Wilczyński [62] are consistent with those obtained here for the PM chronology.

*P. strobus* (PST) is a rare species in Poland that does not form stands that were previously subject to dendroclimatological analyses. A study on the tree ring width–climate relationship in this species was performed in Estonia [63]. Tree ring widths were found to be influenced by the air temperatures in February, March, and September (positive relationships). No statistically significant relationships were obtained for precipitation. In the Czech Republic, however, negative values were obtained for air temperatures of the preceding September in addition to positive correlation values for air temperatures in February and March of the growth year [64]. Statistically significant correlation values also appear in September of the preceding year (positive) and in July and August of the growth year [64]. Correlations from Estonia and the Czech Republic for the previous September and July are consistent with the PST chronology results.

Various studies, models, and forecasts project that climate changes may influence changes in biodiversity, which will increase the threats to the functioning of the existing ecosystems. This may be manifested in shifts to the natural occurrence ranges of tree species and may threaten the Scots pine, i.e., the main forest-forming species. The observed increase in mean annual air temperature and its extremes and lower precipitation in summer, along with frequent periods of drought, further cause an increased risk of fires. The knowledge of the requirements of forest trees with respect to habitat has a high significance in light of the projected environmental changes, especially climatic ones. Direct observations regarding the way trees react to various climatic events (droughts, ground frosts, water relationship changes) influencing their development, growth, and survivability of trees and stands [2,5,9,10] are essential for investigating these requirements. In order to maintain the sustainable development of forest ecosystems, it is essential to monitor changes in climatic conditions, the associated habitat conditions, and their consequences for forests.

## 5. Conclusions

Comparing the values obtained from the study of selected characteristics of six tree species, we note that the introduced plants, *T. plicata* and *P. menziesii*, exceed the native species, *P. sylvestris* and *L. decidua,* both in terms of height and trunk diameter. Also, the obtained growth–climate relationships for these introduced species during the observed increasing winter temperatures and earlier spring beginning indicate improved development conditions for these trees. All tested trees grow in the same rich, fresh forest habitat, which provides them with similar development conditions. In times of dynamic global climate changes, the future use of these species may result in higher wood productivity than domestic species. In the sustainable development of forest ecosystems, knowledge of acclimatization, the tree ring width, and the growth–climate relationship may be useful in the future, especially during climate change, increasing anthropopressure on forests, and a very intense invasion of insect and fungal species attacking native forest-forming species. It is important to continue work on the acclimatization and development of alien species so that their introduction into natural ecosystems does not disturb the ecological balance and biodiversity and leads to sustainable development of the forests.

**Author Contributions:** A.C. and G.N.: conceptualization, methodology, formal analysis, investigation, writing—original draft preparation, writing—review and editing, and visualization. All authors have read and agreed to the published version of the manuscript.

**Funding:** Co-financed by the Minister of Science under the "Regional Excellence Initiative" program for 2024–2027.

**Institutional Review Board Statement:** Not applicable.

**Informed Consent Statement:** Not applicable.

**Data Availability Statement:** Data are available in a publicly accessible repository: Cedro, Anna; Nowak Grzegorz, 2024, "Local chronologies of introduced and native coniferous tree species growing in northwestern Poland (Barlinek Forest Inspectorate)" (https://doi.org/10.18150/CGMPCA).

**Conflicts of Interest:** The authors declare no conflicts of interest.

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
