# Peer review of "A Comparison of the Chronologies of Introduced versus Native Coniferous Tree Species Growing in Northwestern Poland during the Period of Global Warming"

_sustainability, doi:10.3390/su16052215_

Round 1
Reviewer 1 Report
Comments and Suggestions for Authors
Dear authors,
After a careful review of the submitted manuscript, I would like to express my appreciation for the dedicated effort put into the study of chronologies of six species. However, I observe some areas in the text that, in my opinion, require significant improvement before being considered for publication.
Also, throughout the text, within the PDF manuscript file, I highlight several suggestions that are important to take into consideration.
Introduction
The introduction section could be more comprehensive, providing a broader approach. While climate change is a relevant component, it is crucial to avoid it becoming the only explanation or justification for the work. I recommend contextualizing the introduction more extensively, incorporating essential concepts such as ecology, dendroecology, the impact of introduced plant species on the environment, and forest sustainability. Additionally, enrich the introduction with references to previous studies demonstrating significant growth differences among these distinct species and their consequences.
Species Names
While it is acceptable to mention the common/popular names of species in the Materials and Methods, I suggest using scientific names throughout the text. When mentioned for the first time, it is advisable to write the full name of the species, for example, Pinus sylvestris L. (PS). In subsequent references, abbreviations such as P. sylvestris can be used.
Tree Health
The evaluation of tree health was not clear in the hypothesis or Materials and Methods. In the Results, this topic is related to trunk features. I recommend clarifying how this evaluation was conducted and systematizing it, avoiding subjective terms like “good health”, “entirely healthy”, “poorest health”, “weak crowns”, “most healthy”. I also suggest modifying the term "Tree health" to "Tree features" or "Dendrology."
Presentation Order
Maintain the sequence of presenting species in the results and discussion, facilitating the understanding of content and comparisons. Also, standardize how species are mentioned, preferably by their scientific names, or if using abbreviations, define this in the materials and methods.
Objectives and Results
Ensure that objectives, hypotheses, and results are aligned. Some information presented in the results is not among the objectives. Working with hypotheses can strengthen the article, given the comparisons between species and he relationship climate-growth.
Discussion and Conclusion
The presentation of the discussion is confusing. I recommend following the same sequence as the results. Clearly define the objectives to enhance the discussion and conclusion. It is not clear how climate change can affect Poland's forests concerning the studied species. I also suggest comparing introduced and native species, as indicated by the title.
I believe these suggestions will contribute to strengthening the quality of the manuscript. I am available for further clarification and look forward to an improved version of the work.
Sincerely,
Reviewer

Comments on the Quality of English LanguageModerate editing of English language required.
Author Response
SUSTAINABILITY Szczecin, 31.01.2024
Dear Editors and Reviewers,
We are grateful for the insightful analysis of our manuscript, and all the comments, and suggestions provided. We did our best to take into account all the remarks.
We hope that the enclosed revised manuscript meets the requirements of the Editors and Reviewers, and is suitable for publication.
Thank you very much for extending the period to improve the article.
Reviewer 1
We received 4 reviews, sometimes the Reviewers wrote contradictory comments (e.g. to use Latin names/ to use common names or this part needs to be supplemented/extended or this part should be shortened), so it was difficult to fully take into account the comments of all Reviewers. After the discussion, we tried to improve the article as much as possible, taking into account the Reviewers' comments.
All suggestions of Reviewer 1 have been incorporated. Specifically, these are:
Title was changed;
Abstract - Corrected according to the reviewer’s suggestion;
Introduction - Corrected according to the reviewer’s suggestion;
Species Names - we decided use Latin names of species, according to the reviewer’s suggestion [firs full name (Pinus sylvestris), next (P. sylvestris)];
Tree Health - the title of the subsection was changed and its content was adjusted to the reviewers' comments;
Presentation Order - Corrected according to the reviewer’s suggestion;
Objectives and Results - Corrected according to the reviewer’s suggestion;
Discussion and Conclusion - Corrected according to the reviewer’s suggestion;
Line 47-48 - Corrected according to the reviewer’s suggestion;
Lines 58 - Corrected according to the reviewer’s suggestion;
Line 59 - Corrected according to the reviewer’s no. 2 suggestion;
Lines 61 - Corrected according to the reviewer’s suggestion;
Line 97, 100, 104, 109, 113, 118 - Corrected according to the reviewer’s suggestion;
Lines 124 - Corrected according to the reviewer’s suggestion;
Lines 166 - Corrected according to the reviewer’s suggestion; S mean South;
Lines 197- 215, 3.1. Tree health - Corrected according to the reviewer’s suggestion;
Lines 205 - Table 1 is on next page;
Lines 216 - Corrected according to the reviewer’s suggestion;
Lines 217-218, 221, 222 - Corrected according to the reviewer’s suggestion;
Line 231 - Corrected according to the reviewer’s suggestion; we added this objectives;
Line 267 - Corrected according to the reviewer’s suggestion;
Lines 322-326, Figure 4 - Corrected according to the reviewer’s suggestion;
Lines 395-397 - Corrected according to the reviewer’s suggestion;
Line 399 - Corrected according to the reviewer’s suggestion;
Lines 405 - Corrected according to the reviewer’s suggestion;
Lines 459-461 - Corrected according to the reviewer’s suggestion;
Lines 470-493, Conclusions - Corrected according to the reviewer’s suggestion;
We are grateful for the insightful analysis of our manuscript, and all the comments, and suggestions provided. We did our best to take into account all the remarks. We hope that the enclosed revised manuscript meets the requirements of the Editors and Reviewers, and is suitable for publication.
Research data will be placed in an open repository after the article is accepted for publication.
We hope that the Reviewers and Editors find the current form of the article acceptable for publication in this journal.
Sincerely,
the authors
Anna Cedro and Grzegorz Nowak

Reviewer 2 Report
Comments and Suggestions for Authors
Specific comments:
1. Lines 13-33 of the Abstract section, it is necessary to eliminate certain irrelevant information such as lines 22-26 and 29-30; this type of fundamental details can be relocated to section 2.1 Study Area. Additionally, the existing findings on the growth rates of three tree species and their relationships with climate appear excessively brief and devoid of statistical analysis.
2. Line 29. Please use italic writing for "Thuja plicata".
3. Lines 38-57, the Introduction, need to be rewritten and well organized. The authors appear to emphasize the significant importance of conducting this study without mentioning any recent or relevant research advancements.
4. Lines 63-66. Besides the location, tree species composition, and geosoil formation, it is also important to introduce the long-term climate variations such as air temperature and precipitation.
5. Lines 71-87. Irrelevant information can be eliminated here.
6. Lines 90-122. In order to effectively compare the selected six tree species, it is suggested to provide a table that includes closely related characteristics after the text. Once the authors have provided such a table (I see Table 1 later), then the corresponding text can be further simplified.
7. Line 106. One more space was found between "the" and "region". Please carefully check throughout the manuscript, authors. Also, please see line 126.
8. Line 147. The term "EPS" should be defined in the main text instead of being mentioned only in the titles of tables, figures, or notes.
9. Lines 161-162, in Table 1. The authors carefully corrected the mistake that occurred in the column name (tree-rings) of the last column in the table. Please also refer to line 188 for the temperature unit.
10. Line 166. What does the "S" stand for after the "22-25 km"?
11. Another section on how to conduct data analysis should be included in "2. Materials and Methods".
12. Generally, the authors should reorganize the section of '2. Materials and Methods' by arranging the related parts together to facilitate readers' understanding.
13. Lines 192-195, in Figure 2. Why not use the months from January to December instead of I to XII? The usage of Roman numerals may lead to confusion later on and is rarely seen. Additionally, it was suggested to include the variation (standard deviation) for P, T, and IN in the figure.
14. Lines 197-215. The current status of this part of the result, namely tree health, was not adequately supported. The authors should provide indicators for evaluating tree health and quantify these indicators instead of merely describing general tree characteristics such as height, trunk diameter, natural branches, rotten stem trunks, and withered trees through text without sufficient figure data or statistical analysis.
15. Lines 225-227. The TP chronology appears twice in these sentences, so please carefully check it.
16. Lines 258-261, in Table 2. The year 2023 had just passed, so how could the authors obtain the tree-ring width in 2023 as shown in the Time Span and EPS series? Did they even obtain a time span ranging from 1899 to 2025?
17. Lines 262-265, in Figure 3. The variation (standard deviation) in each of the six tree species was suggested to be added to the figure. By the way, was the displayed figure obtained from the data after detrending for the influence of long-term tree ages?
18. Lines 322-327, in Figure 4. What is the variable name for the y-axis? Is it necessary to both display the correlation coefficient and response function (the slope of the linear function, because the coefficients include positive and negative values? In other words, what did the response function mean?)?
19. Line 338. Please denote which figure it (Figure X) refers to, see line 360 as well.
20. When describing the analysis of the pointer years, why do the authors only focus on the similarity reactions based on the 3 out of 6 chronologies? What is the significance of this operation, or are there any other studies performed as reference? As we all know, different tree species could show vary responses to the same climate factors, so that the authors had better analyze the individual tree species’ growth in response to climate, rather than most of the six tree species. After performing this analysis, the vary responses of tree-ring width to climate will shed light to fight climate change, to reveal which tree species grew fast and healthy for carbon sink and timber, and was more adaptive under the context of the global climate change.
21. Lines 341-356, 362-375. These sentences the authors provided here seem to try to explain the reasons why the positive/negative pointer years occurred based on the analysis regarding the relationships between the tree-ring widths and winter/early spring air temperature, summer precipitation, etc. Anyway, no matter the reasons were reliable or not, these parts should move to the part of the discussion.
22. Lines 382-387, Figure 5. The figure should be further beautified.
23. Lines 398-399. The reasons the authors try to explain the wide tree-ring width, in building the connection to the winter and early spring T, and high P, as well as low IN, is feeble and lack of supporting evidence from similar studies. In other words, the data analysis should be further explored to strengthen the evidence.
24. See my comment 13. As discussing the reasons why the tree-ring width shows significant relations to winter/early T and summer P, the sequence for the annual month is a big confusing thing that is hard to transform from I, ... XII to JAN, ... DEC.
25. As shown in Figure 4, the tree-ring width for PS, CL, TP, and PM was positively related to T mainly in FEB, partially extended to APR. The above is the similarity in T, however, the positive relation for P move to May (LD, CL, TP, and PM), of which month belongs to later spring or early summer. These results seem to differ from the conclusion drawn by the authors. Furthermore, the preceding year (pSEP, pOCT, pNOV, pDEC) and the vegetation year (FEB, MAR, MAY, and JUL) of P seem positive to TRW among at least five tree species!!! What’s more, as denoted by my comment 20!!! Besides finding out the similarity, the correlation and response function of the individual tree species to the climate were much more important, as it indicates the lights for the responses of tree growth of each tree species to the global change, i.e., the increased temperature, and the intensified and frequent summer drought.
26. Lines 382-387, and 403-404, 410-412, and 453-455, in Figure 5, and it is related discussion. The authors denote the pointer year when 3 out of 6 tree shown similar growth trends, but the readers have no idea about the detailed climate of the designated positive/negative pointer years, how can such a kind of explanation support the views which the authors raised and make the readers believe?
Lines 470-493, the Conclusions. After addressing the whole issues raised above, the conclusions should be redrawn, and the current one is not well organized.
Comments on the Quality of English LanguageThe formats of this manuscripot consist lots of problems.
Author Response
Reviewer 2
We received 4 reviews, sometimes the Reviewers wrote contradictory comments (e.g. to use Latin names/ to use common names or this part needs to be supplemented/extended or this part should be shortened), so it was difficult to fully take into account the comments of all Reviewers. After the discussion, we tried to improve the article as much as possible, taking into account the Reviewers' comments.
All suggestions of Reviewer 2 have been incorporated. Specifically, these are:
Specific comments:
- Lines 13-33- Corrected according to the reviewer’s suggestion;
- Line 29 - Corrected according to the reviewer’s suggestion;
- Lines 38-57 - Corrected according to the reviewer’s suggestion;
- Lines 63-66 - Corrected according to the reviewer’s suggestion; next subchapter 2.2. describes the climate of this region;
- Lines 71-87 - Corrected according to the reviewer’s suggestion;
- Lines 90-122 - In our opinion, a brief characterization of the species studied is necessary;
- Line 106 - Corrected according to the reviewer’s suggestion;
- Line 147 - Corrected according to the reviewer’s suggestion;
- Lines 161-162, 188 - Corrected according to the reviewer’s suggestion; thank you very much for pointing out this error;
- Line 166 - Corrected according to the reviewer’s suggestion; S mean South;
- Lines 164-195 (section Climate Data) was removed - Corrected according to the reviewer’s suggestion;
- Corrected according to the reviewer’s suggestion;
- Lines 192-192, Figure 2 - Corrected according to the reviewer’s suggestion; We include two Figure 2 in the article (the choice is made by the Reviewer and the Editor). In our opinion, adding a standard deviation to the T, P and IN series (additional 6 lines) makes the drawing unreadable;
- Lines 197-215 - Corrected according to the reviewer’s suggestion;
- Lines 225-227 - this is not an error, this is how it should be, please see Figure 3. 16. Lines 258-261 - Corrected according to the reviewer’s suggestion; thank you very much for pointing out this error; Pinus sylvestris and Pinus strobus were sampling in November 2023, when vegetation season 2023 was finished, we added this information in section Tree-Ring Data;
- Lines 262-265, Figure 3 - In our opinion, adding a standard deviation to 6 series (additional 12 lines) makes the drawing unreadable; no this data are tree-ring width in mm (before detrending);
- Lines 322-327, Figure 4 - Corrected according to the reviewer’s suggestion; we added new Figure 4;
- Linea 338, 360 - Corrected according to the reviewer’s suggestion; thank you very much for pointing out this error;
- In our opinion, the analysis of pointer years is more suitable for showing common features due to the similarity of chronologies and the growth-climate relationship (in our opinion, response function analysis is better for showing differences).Therefore, we presented common pointer years for several species.We have also added a table with weather characteristics in these years. - Corrected according to the reviewer’s suggestion;
- Lines 341-356, 362-375 - We do not provide the reasons here, but an analysis of the weather conditions in these years, this, in our opinion, is part of the results, not a discussion
- Lines 382-387, Figure 5 - Corrected according to the reviewer’s suggestion;
- Lines 398-399 - These are our results (analysis of response functions and pointer years) and those of many other authors, as we show in the discussion
- Corrected according to the reviewer’s suggestion;
- Corrected according to the reviewer’s suggestion;
- Lines 382-387, 403-404, 410-412, 453-455, Figure 5 - we added Table 4 with weather characteristics in pointer years.
Corrected according to the reviewer’s suggestion;
- Lines 470-493 - Corrected according to the reviewer’s suggestion;
We are grateful for the insightful analysis of our manuscript, and all the comments, and suggestions provided. We did our best to take into account all the remarks. We hope that the enclosed revised manuscript meets the requirements of the Editors and Reviewers, and is suitable for publication.
We hope that the Reviewers and Editors find the current form of the article acceptable for publication in this journal.
Research data will be placed in an open repository after the article is accepted for publication.
Sincerely,
the authors
Anna Cedro and Grzegorz Nowak

Reviewer 3 Report
Comments and Suggestions for Authors
Notes:
1. The title of the article provides “Comparison of the width of tree rings in introduced and domestic species of coniferous trees ...”, but in the abstract the comparison is limited only to the average width of the tree ring.
2. The purpose of the study by the authors is “to study the growth rate and health of trees of various coniferous species...”. The annotation contains no information at all about the health of the trees. The text of the article (section 3.1) provides information only on visual inspection of trees to assess health. It is obvious that quantitative analysis with the establishment of trial plots and the use of generally accepted silvicultural methods was not used.
The authors only indicated that the sampling was carried out from “the healthiest trees”.
3. The relationship between the width of tree rings and climate in the abstract and in the conclusions is not differentiated by the species of trees studied, although this information is available in the text of the article (sections 3.3 and 4).
4. The article contains a general description of the forest conditions of the site, but does not indicate the features (or lack thereof) in the places where samples were taken for research. The authors noted a large difference in heights at the research site, which could also affect the average width of the increments of the selected samples.
5. There is a typo in the header of Table 1, last column: tee rings (should - tree rings).
6. There is a typo in Table 2, the last line: the year 2025 is indicated in the “Time Span” column.
Author Response
Reviewer 3
We received 4 reviews, sometimes the Reviewers wrote contradictory comments (e.g. to use Latin names/ to use common names or this part needs to be supplemented/extended or this part should be shortened), so it was difficult to fully take into account the comments of all Reviewers. After the discussion, we tried to improve the article as much as possible, taking into account the Reviewers' comments.
All suggestions of Reviewer 3 have been incorporated. Specifically, these are:
Notes:
1: The title of the article has been changed. Corrected according to the reviewer’s suggestion.
2: The title of the subsection was changed and its content was adjusted to the reviewers' comments; Corrected according to the reviewer’s suggestion.
3: In the comments of other reviewers there were suggestions to change the abstract and conclusions, which was done
4: There are no big differences in the height above sea level of the examined trees (only a dozen or so meters), but there are differences in the height of the trees - these are given in Table 1.
5: Corrected according to the reviewer’s suggestion; thank you very much for pointing out this error.
6: Corrected according to the reviewer’s suggestion; thank you very much for pointing out this error.
We are grateful for the insightful analysis of our manuscript, and all the comments, and suggestions provided. We did our best to take into account all the remarks. We hope that the enclosed revised manuscript meets the requirements of the Editors and Reviewers, and is suitable for publication.
Research data will be placed in an open repository after the article is accepted for publication.
We hope that the Reviewers and Editors find the current form of the article acceptable for publication in this journal.
Sincerely,
the authors
Anna Cedro and Grzegorz Nowak

Reviewer 4 Report
Comments and Suggestions for Authors
I attach an annotated pdf to help authors revising the ms. Among these editing comments I suggest changing the title. Please be also very consistent with the names of species. I suggest using the common names throughout the ms.
The Discussion and Conclusions are mainly a repetition of results. They should be DEEPLY revised to show readers if climate warming will improve more the growth of introduced vs native conifers, etc. You should also provide references (e.g., from ecophysiology) to explain why some species respond more to some climate factors (e.g. warm springs) than others. Discuss on cold and drough tolerance of the compared conifers!

Comments on the Quality of English LanguageThe english needs some minor/moderate editing in some parts of the ms. I suggest making more simple sentences.
Author Response
Reviewer 4
We received 4 reviews, sometimes the Reviewers wrote contradictory comments (e.g. to use Latin names/ to use common names or this part needs to be supplemented/extended or this part should be shortened), so it was difficult to fully take into account the comments of all Reviewers. After the discussion, we tried to improve the article as much as possible, taking into account the Reviewers' comments.
The attached pdf file contains instructions that we often do not understand (there are deletions in red, deletions in blue and text marked in yellow). We tried to read the Reviewer's intentions, but it was very difficult for us (e.g., line 147 or the word Convergence, should we replace it with another expression or remove the entire sentence, or line 267, caption Table 3, should we remove or replace the word Convergence?)
All suggestions of Reviewer 4 have been incorporated. Specifically, these are:
Title was changed;
Reviewer 1 suggested using Latin species names and so we changed these names throughout the article;
Discussion and Conclusions were changed; Corrected according to the reviewer’s suggestion;
We are grateful for the insightful analysis of our manuscript, and all the comments, and suggestions provided. We did our best to take into account all the remarks. We hope that the enclosed revised manuscript meets the requirements of the Editors and Reviewers, and is suitable for publication.
Research data will be placed in an open repository after the article is accepted for publication.
We hope that the Reviewers and Editors find the current form of the article acceptable for publication in this journal.
Sincerely,
the authors
Anna Cedro and Grzegorz Nowak

Round 2
Reviewer 1 Report
Comments and Suggestions for Authors
The authors have completed the review of the manuscript, and the suggested changes have been implemented, improving the quality of the article.
Author Response
SUSTAINABILITY Szczecin, 28.02.2024
Dear Editors and Reviewers,
Reviewer 1
Thank you very much for reading our article again and comments. Manuscript revision has been performed. They were included in the attached version of the article.
Research data will be placed in an open repository after the article is accepted for publication.
We hope that the Reviewers and Editors find the current form of the article acceptable for publication in this journal.
Sincerely,
the authors
Anna Cedro and Grzegorz Nowak

Reviewer 2 Report
Comments and Suggestions for Authors
I carefully read the reply made by the authors. Some of my comments were not well addressed, in others words, I don't satisfy with the reply drawn by the authors since the authors just reply "Corrected according to the reviewer’s suggestion;" to me, in a way of avoiding my major concerns. Furthermore, as revised Figures 2 and 3, the authors can add error bar to the figure rather than drawing series of lines instead... ...
Author Response
SUSTAINABILITY Szczecin, 19.02.2024
Dear Editors and Reviewers,
We are grateful for the insightful analysis of our manuscript, and all the comments, and suggestions provided. We did our best to take into account all the remarks.
We hope that the enclosed revised manuscript meets the requirements of the Editors and Reviewers, and is suitable for publication.
Reviewer 2
Thank you very much for reviewing again and taking the time to improve our text. We apologize for often providing such laconic answers (”Corrected according to the reviewer’s suggestion”), but it does not mean that we did not pay attention to them. However, with four full reviews, there were so many comments and they often covered the same parts of the text and were sometimes contradictory, which made it difficult to correct and respond to these comments.
Specific comments:
- Lines 13-33 - Corrected according to the reviewer’s suggestion; Abstract: we have removed information indicated as unnecessary by the reviewer, we have added information indicated by the reviewer (regarding the growth-climate relationship);
- Line 29 - Corrected according to the reviewer’s suggestion; We used italic writing for all species names;
- Lines 38-57 - Corrected according to the reviewer’s suggestion; We tried to change the introduction so that the Reviewer's comments were taken into account, and new literature items were added;
- Lines 63-66 - Corrected according to the reviewer’s suggestion; next subchapter 2.2. describes the climate of this region;
- Lines 71-87 - Corrected according to the reviewer’s suggestion; we have removed information indicated as unnecessary by the reviewer;
- Lines 90-122 - In our opinion, a brief characterization of the species studied is necessary;
here we compare the listed species growing in natural habitats, and the table shows the tested trees;
- Line 106 - Corrected according to the reviewer’s suggestion; the space has been removed;
- Line 147 - Corrected according to the reviewer’s suggestion; the term EPS has been clarified;
- Lines 161-162, 188 - Corrected according to the reviewer’s suggestion; thank you very much for pointing out this error;
- Line 166 - Corrected according to the reviewer’s suggestion; S mean South; was changed to south and 23 km;
- Lines 164-195 (section Climate Data) was removed - Corrected according to the reviewer’s suggestion;
- Corrected according to the reviewer’s suggestion; sections 2. Material and methods have been reorganized;
- Lines 192-192, Figure 2 - Corrected according to the reviewer’s suggestion; We include two Figure 2 in the article (the choice is made by the Reviewer and the Editor). In our opinion, adding a standard deviation to the T, P and IN series (additional 6 lines) makes the drawing unreadable; We have corrected Figure 2 according to the reviewer's comments, so the error bars will, in our opinion, make the image unreadable too. Another reviewer suggests keeping version A of this Figure, we leave the decision to the Editors;
Please see (examples of figures showing temperature and precipitation or other meteorological factors for the study area, without STD):
Forests 2024, 15(2), 355; https://doi.org/10.3390/f15020355
Forests 2024, 15(2), 317; https://doi.org/10.3390/f15020317
Forests 2024, 15(2), 345; https://doi.org/10.3390/f15020345
Forests 2024, 15(2), 243; https://doi.org/10.3390/f15020243
Sustainability 2023, 15(24), 16755; https://doi.org/10.3390/su152416755
Sustainability 2021, 13(20), 11376; https://doi.org/10.3390/su132011376
- Lines 197-215 - Corrected according to the reviewer’s suggestion; We agree with the comment that the description does not concern the health of trees, therefore it was changed to a description of tree features - the title of the subsection, the description was also changed;
- Lines 225-227 - this is not an error, this is how it should be, please see Figure 3.
- Lines 258-261 - Corrected according to the reviewer’s suggestion; thank you very much for pointing out this error; Pinus sylvestris and Pinus strobus were sampling in November 2023, when vegetation season 2023 was finished, we added this information in section Tree-Ring Data;
- Lines 262-265, Figure 3 - In our opinion, adding a standard deviation to 6 series (additional 12 lines) makes the drawing unreadable; no this data are tree-ring width in mm (before detrending); besides, if we add the STD for TRW, with a scale on the vertical axis of up to 400 mm, these data are completely invisible (e.g. for the PS chronology, the STD is 0.922 mm - Table 2, after adding and subtracting this value from each value from the PS series and drawing it this is not visible in Figure 3). We tried to show the STD, but in our opinion it doesn't add anything new, it just obscures the picture. I don't understand how to show STD for accumulated series, please give me an example of what it should look like according to the Reviewer.
example for one PS series
Please see (examples of figures showing cumulative radial growth, without STD):
- Plant Ecology 212(7):1123-1134, DOI: 1007/s11258-010-9892-9;
- Journal of the Royal Statistical Society Series C: Applied Statistics, 2023, 00, 1–20; https://doi.org/10.1093/jrsssc/qlad015;
· Dendrochronologia, Volume 49, June 2018, Pages 89-93, https://doi.org/10.1016/j.dendro.2018.03.003;
-
Trees (2012) 26:283–290, DOI 10.1007/s00468-011-0590-6.
- Lines 322-327, Figure 4 - Corrected according to the reviewer’s suggestion; we added new Figure 4;
- Linea 338, 360 - Corrected according to the reviewer’s suggestion; thank you very much for pointing out this error;
- In our opinion, the analysis of pointer years is more suitable for showing common features due to the similarity of chronologies and the growth-climate relationship (in our opinion, response function analysis is better for showing differences).Therefore, we presented common pointer years for several species.We have also added a table with weather characteristics in these years. - Corrected according to the reviewer’s suggestion;
- Lines 341-356, 362-375 - We do not provide the reasons here, but an analysis of the weather conditions in these years, this, in our opinion, is part of the results, not a discussion;
- Lines 382-387, Figure 5 - Corrected according to the reviewer’s suggestion; colors, contrast and character size have been changed;
- Lines 398-399 - These are our results (analysis of response functions and pointer years) and those of many other authors, as we show in the discussion ;
- Corrected according to the reviewer’s suggestion; the names of the months I, II, ..., XII were replaced in the Figures and throughout the text with JAN, FEB, ..., DEC or January, February....;
- Corrected according to the reviewer’s suggestion; Fig. 4 has been completely revised following reviewer comments and the results (entire description) have also been revised;
- Lines 382-387, 403-404, 410-412, 453-455, Figure 5 - we added Table 4 with weather characteristics in pointer years. Corrected according to the reviewer’s suggestion;
- Lines 470-493 - Corrected according to the reviewer’s suggestion; the conclusions were rewritten taking into account the reviewer's comments;
We are grateful for the insightful analysis of our manuscript, and all the comments, and suggestions provided. We did our best to take into account all the remarks. We hope that the enclosed revised manuscript meets the requirements of the Editors and Reviewers, and is suitable for publication.
We hope that the Reviewers and Editors find the current form of the article acceptable for publication in this journal.
Sincerely,
the authors
Anna Cedro and Grzegorz Nowak

Reviewer 4 Report
Comments and Suggestions for Authors
The ms can be accepted given the adequate revision performed. Regarding Fig 2, I suggest keeping the A version.
Author Response
SUSTAINABILITY Szczecin, 19.02.2024
Dear Editors and Reviewers,
We are grateful for the insightful analysis of our manuscript, and all the comments, and suggestions provided. We did our best to take into account all the remarks.
We hope that the enclosed revised manuscript meets the requirements of the Editors and Reviewers, and is suitable for publication.
Reviewer 4
Thank you very much for reading our article again and comments. Manuscript revision has been performed. They were included in the attached version of the article.
Research data will be placed in an open repository after the article is accepted for publication.
We hope that the Reviewers and Editors find the current form of the article acceptable for publication in this journal.
Sincerely,
the authors
Anna Cedro and Grzegorz Nowak

Round 3
Reviewer 2 Report
Comments and Suggestions for Authors
In Figure 2, please at least add another two lines for temperature, Tmin and Tmax, because they also can help show the variations in temperature, which were important for the determination of tree growth. See https://doi.org/10.1016/j.gecco.2023.e02743 https://doi.org/10.1016/j.dendro.2022.126023
Author Response
SUSTAINABILITY Szczecin, 28.02.2024
Dear Editors and Reviewers,
Reviewer 2
Figure 2 - Corrected according to the reviewer’s suggestion; Tmax and Tmin distribution have been added.
Research data will be placed in an open repository after the article is accepted for publication.
We hope that the Reviewers and Editors find the current form of the article acceptable for publication in this journal.
Sincerely,
the authors
Anna Cedro and Grzegorz Nowak
